# Electromagnetic Interference Effects of Continuous Waves on Memristors: A Simulation Study

**DOI:** 10.3390/s22155785

**Published:** 2022-08-03

**Authors:** Guilei Ma, Menghua Man, Yongqiang Zhang, Shanghe Liu

**Affiliations:** 1National Key Laboratory on Electromagnetic Environment Effects, Shijiazhuang Campus, Army Engineering University, Shijiazhuang 050003, China; mgljyp@163.com (G.M.); zyq@hebust.edu.cn (Y.Z.); 2School of Information Science and Engineering, Hebei University of Science and Technology, Shijiazhuang 050018, China

**Keywords:** electromagnetic interference effects, memristor, neuromorphic system, memristor modeling, reliability, continuous wave

## Abstract

As two-terminal passive fundamental circuit elements with memory characteristics, memristors are promising devices for applications such as neuromorphic systems, in-memory computing, and tunable RF/microwave circuits. The increasingly complex electromagnetic interference (EMI) environment threatens the reliability of memristor systems. However, various EMI signals’ effects on memristors are still unclear. This paper selects continuous waves (CWs) as EMI signals. It provides a deeper insight into the interference effect of CWs on the memristor driven by a sinusoidal excitation voltage, as well as a method for investigating the EMI effect of memristors. The optimal memristor model is obtained by the exhaustive traversing of the possible model parameters, and the interference effect of CWs on memristors is quantified based on this model and the proposed evaluation metrics. Simulation results indicate that CW interference may affect the switching time, dynamic range, nonlinearity, symmetry, time to the boundary, and variation of memristance. The specific interference effect depends on the operating mode of the memristor, the amplitude, and the frequency of the CW. This research provides a foundation for evaluating EMI effects and designing electromagnetic protection for memristive neuromorphic systems.

## 1. Introduction

The memristor was predicted to be the fourth fundamental passive circuit element (in addition to the resistor, capacitor, and inductor) in the world by Leon Chua in 1971 [1]. There has been an increasing interest in memristors and their applications since the real memristor was implemented at Hewlett-Packard Labs [2,3]. Many memristors have recently been implemented based on different device concepts. These include phase change memory (PCM), conductive bridging random access memory (CBRAM), resistive random access memory (RRAM), spintronic devices, ferroelectric devices, the self-directed channel (SDC) device [4,5], and others. Moreover, due to their low power consumption, nanometer size, nonvolatile characteristics, good scalability, and compatibility with CMOS technology [6,7,8], memristors have been implemented in many applications, such as neuromorphic computing [9,10,11], in-memory computing [12], combinational logic circuits [13,14], spintronic devices, and hardware security systems [15]. In recent years, in pursuit of reconfigurability and tunability, several studies have applied memristors to RF/microwave circuits and a broader context in electromagnetic systems [16], such as frequency selective surfaces [17], the reconfigurable antenna [17], R.F./microwave filter [18,19,20,21], and Wilkinson power divider [22]. As research on memristors advances, new memristive applications are continuously emerging.

It is worth noting that with the advancement of electron science and technology, the number of radio-using devices is increasing, the space is filled with various intentional and unintentional electromagnetic wave signals that ramify and overlap in the time domain, frequency domain, and space domain [23], which are serious threats to the reliability of highly integrated and miniaturized memristive systems. Previous studies have focused on the reliability issues of memristors, including endurance, retention deterioration, read/write noise [24], and signal integrity issues [25,26,27,28], which may be caused by manufacturing processes, read/write operations, structure, or size of the memristors. However, the interference effects of external EMI on memristors have rarely been considered. Under the threat of the increasingly complex electromagnetic environment, EMI signals may interfere with the characteristics of the memristor system by coupling into the memristive system through wires or holes, thereby affecting the performance of the memristive system. Therefore, it is essential to investigate the interference effects of EMI signals on memristors to achieve high-performance memristive applications.

In this study, continuous waves (CWs) are selected as the EMI signals. Their interference effect on the memristor is examined exhaustively utilizing the Knowm self-directed channel (SDC) memristor as the example for the first time. The Knowm SDC memristor is the first commercially available memristor fabricated by Knowm Inc. (Santa Fe, NM, USA) and is based on a similar electrochemical metal mechanism as CBRAM devices to change the device conductivity. However, the SDC device constrains the movement of metal ions to a chemically created channel structure, resulting in a much more consistent and stable device [3,4]. This research paper covers two main aspects. On the one hand, an optimized memristor model is developed based on the experimental data of Knowm SDC by exhaustively traversing the possible model parameters. On the other hand, the interference effect of CWs on the memristor under sinusoidal excitation was examined utilizing suggested evaluation metrics, confirming that the memristor is not subject to phase change and permanent damage. It is intended that the study presented in this paper will give insight into the interference effects of CWs on memristors and contribute to the evaluation of electromagnetic effects and electromagnetic protection design of memristive systems.

The rest of the paper is organized as follows: the experimental Knowm SDC memristors, their modeling methods, and evaluation metrics of the EMI effect are described in Section 2. Simulation results and discussion are presented in Section 3, followed by concluding remarks in Section 4.

## 2. Components and Methods

### 2.1. The Knowm SDC Memristor

The device studied in this work is the Knowm SDC memristor, a BS-AF-W discrete bipolar memristor with tungsten (W) dopant on a chalcogenide material, developed and commercialized in a 16-pin ceramic DIP package by Knowm Inc. It operates primarily through the electric field-induced generation of metal ions that move through a multilayer chalcogenide material stack to change the memristance [3,4,29]. Compared with the conventional mechanism of using oxygen vacancy migration to change the memristance, the means of changing the memristance in the Knowm SDC memristor improves the stability of the memristor’s electrical performance [30].

Figure 1 presents the W-dopant SDC memristor device structure, switching mechanism, and symbol. The memristor device consists of two electrode layers, a multilayer chalcogenide material stack, and an Ag-source layer. During the first run after device fabrication, when a positive potential is applied to the top electrode, metal ions reach the lower potential electrode with the assistance of Sn^+^ from the SnSe layer and are reduced to the metallic form, eventually forming a conductive channel spanning the active material layer. Then, by applying the potentials of different polarities to the memristor, its memristance changes. When a positive potential is applied to the memristor, Ag^+^ is forced to move in the direction of the electric field to the conductive channel and accumulate at the aggregation sites, resulting in a decrease in the memristance. Conversely, when a negative potential is applied to the memristor, the memristance is tunable in the higher direction by the movement of Ag^+^ away from these agglomeration sites. Its maximal and minimal memristances are known as the OFF resistance (*R*_OFF_) and the ON resistance (*R*_ON_), respectively. The lowest and highest resistance states are called LRS and HRS, respectively, and the other medium resistance states are all called MRS [24].

Depending on the dynamic range of the memristance, the memristor operates in hard-switching mode (H-Sw mode) if the dynamic range is between *R*_ON_ and *R*_OFF_. In contrast, the memristor operates in soft-switching mode (S-Sw mode) [6] if one boundary of the dynamic range is at MRS.

### 2.2. Modeling of the Knowm SDC Memristor

Before modeling the Knowm SDC memristor, the I-V characteristics must be experimentally determined. Then, the I-V characteristics are fitted using a generic memristor model, and the parameters of the generic memristor model are optimized by exhaustive traversal of all parameters to achieve the best fit; subsequently, the optimal model parameters are identified. Finally, the generic memristor model with the optimal parameters can be utilized as the memristor model.

#### 2.2.1. Experiment Setup of the SDC Memristor

According to Knowm’s recommendation, the applied voltage and the device current should be limited to less than 1 V and 1 mA [29], respectively; otherwise, excessive voltages or currents can induce phase change or permanent burnout of the memristor. Therefore, a linear resistor *R_s_* in series with the memristor is required in the experimental connection, as shown in Figure 2. The voltage across *R_s_* and the sinusoidal excitation applied to the memristor M1 can be monitored by channel CH1 and channel CH2 of the oscilloscope and noted as *V_s_* and *V_in_* respectively. Then the voltage *V_m_* across M1 is *V_in_*-*V_s_*, and the current *I_m_* flowing through M1 is *V_s_*/*R_s_*, leading to the memristor’s I-V characteristic.

#### 2.2.2. The Generic Memristor Model

Previous studies have attempted to model the Knowm W-dopant SDC memristor [32,33] by optimizing parameters of generic memristor models, and the best approximation was obtained from a voltage threshold adaptive memristor model (VTEAM [34]). However, the fitting error was significant even in this case. The mean metastable switch (MMS) memristor model [35] considered in this work is a variant of the metastable switch memristor model, which is a semi-empirical generality model jointly proposed by M. Nugent and T. Molter [36,37]. It arises from the notion that memristors can be represented as a collection of conducting channels that switch between states of differing resistance. Each conducting channel is represented by a metastable switch.

A metastable switch is an idealized two-state element that can probabilistically switch between two states under the influence of voltage bias and temperature. The probability of the metastable switch switching from the OFF to the ON state is denoted by *P*_ON_, while the probability of the metastable switch switching from the ON to the OFF state is denoted by *P*_OFF_. The switching probabilities are modeled as:(1)$$P_{\mathrm{ON}}=\alpha \frac{1}{1+e^{-\beta (v-V_{\mathrm{ON}})}}$$
(2)$$P_{\mathrm{OFF}}=\alpha (1-\frac{1}{1+e^{-\beta (v+V_{\mathrm{OFF}})}})$$
where *V*_ON_ is the switching threshold voltage for the ON state and *V*_OFF_ is the switching threshold voltage for the OFF state. $\beta =q/kT=V_{T}{}^{-1}$ is the temperature parameter, *q* is the elementary charge, *k* is the Boltzmann constant, *T* is the absolute temperature, α = dt/τ is the time parameter, and τ is the time constant of the memristor.

The change in the number of switches *x*, scaled from 0 to 1, is defined as:(3)$$dx=P_{\mathrm{ON}}(1-x)-P_{\mathrm{OFF}}x$$

By substituting Equations (1) and (2) into (3), the equation for the state variable of the memristor takes the form of:(4)$$\frac{dx}{dt}=\frac{1}{\tau }(\frac{1}{1+e^{-\beta (v-V_{\mathrm{ON}})}}(1-x)-(1-\frac{1}{1+e^{-\beta (v+V_{\mathrm{OFF}})}})x)$$

Furthermore, the conductivity of the memristor is as follows:(5)$$G_{m}(x)=\frac{x}{R_{\mathrm{ON}}}+\frac{1-x}{R_{\mathrm{OFF}}}$$

As seen in Equations (4) and (5), the behavior of the memristor can be determined by five parameters *R*_ON_, *R*_OFF_, *V*_ON_, *V*_OFF_, and the initial state *x*_0_. In addition, with a known initial memristance, *x*_0_ can be obtained according to Equation (5).

In this article, the MMS memristor model is implemented in MATLAB, and the program code for the MMS model is available in Appendix A.

#### 2.2.3. Optimizing the Parameters of the Generic Memristor Model

This section aims to identify the optimal parameters for the MMS memristor model to fit the SDC memristor I-V characteristics so that the relative sum of squared errors (*rSSE*) is minimized. The *rSSE* is derived from the relative RMS error from [34] and is not too small compared to the relative RMS error, thus highlighting the differences between the fitted results for each group of parameters. The smaller the rSSE, the better the MMS memristor model fits the SDC memristor I-V characteristics. The rSSE is defined as:(6)$$\mathrm{rSSE}=\frac{SSE_{V}}{{\bar{V}}_{exp}^{2}}+\frac{SSE_{I}}{{\bar{I}}_{exp}^{2}}=\frac{\sum_{i=1}^{n}{\left(V_{sim,i}-V_{exp,i}\right)}^{2}}{\sum_{i=1}^{n}V_{exp,i}^{2}}+\frac{\sum_{i=1}^{n}{\left(I_{sim,i}-I_{exp,i}\right)}^{2}}{\sum_{i=1}^{n}I_{exp,i}^{2}}$$
where *n* is the number of samples. $V_{sim,i}$ and $I_{sim,i}$ are the corresponding *i*th sample of the voltage and current of the MMS model, respectively. $V_{exp,i}$ and $I_{exp,i}$ are the corresponding *i*th sample of the experimental voltage and current of the SDC memristor, respectively. ${\bar{V}}_{exp}^{2}$ and ${\bar{I}}_{exp}^{2}$ are the Euclidean norms of the experimental voltage and current of the SDC memristor, respectively.

The procedures for optimizing the model parameters are as follows: first, the range of MSS memristor model parameters, including *R*_ON_, *R*_OFF_, *V*_ON_, *V*_OFF_, and *x*_0_ is estimated based on the experimentally obtained I-V characteristic curve of the SDC memristor; then, all possible parameters within the ranges are exhaustively traversed; finally, the set of parameters with the smallest *rSSE* is taken as the optimal set of parameters for the MSS memristor modeling SDC memristor.

### 2.3. Simulation of the EMI Effect of Memristors

#### 2.3.1. Simulation Setup

In order to examine the interference effects of CW of varying amplitudes and frequencies on memristors with different operating modes, the interference effects of memristors investigations in this part are based on the optimized memristor model presented in Section 2.2. Simulating the coupling of CWs at the memristor input, the CWs were superimposed directly onto the sinusoidal excitation voltage. Then, simulations are performed to record the I-V characteristics and the memristance under the signal stimulation containing CW interference. Finally, the interference effect of CW on the memristor is quantified based on the evaluation metrics proposed in Section 2.3.2. Moreover, to prevent phase change and permanent destruction of the memristor after CW interference, the amplitudes (Amps) of CWs were limited in the range [0.05, 0.5] V when the memristor was operating in H-Sw mode and in the range [0.05, 0.7] V when it was operating in S-Sw mode, according to the voltage and current limits specified by Knowm Inc. For all modes of operation, the CW frequencies (Freqs) fell within the range [1k, 1M] Hz. The amplitude step for the simulation was 0.01 V. Since the interference effect of higher frequency CWs on the memristor was almost the same, the simulation step of frequency depended on the frequency range in order to shorten the simulation time or reduce the repetitive data. The simulation step of frequency was larger at the higher frequency range, while the simulation step of frequency was smaller at the lower frequency range. For example, the simulation step for frequencies in the lower frequency range [1k, 10k] Hz was 100 Hz, while the simulation step for frequencies in the higher frequency range [500k, 1M] was 50 kHz.

#### 2.3.2. Evaluation Metrics of EMI Effects for the Memristor

In this article, the *rSSE* between the memristor response with and without interference is used to evaluate the interference intensity of CWs on the memristor (Table 1). The larger the *rSSE*, the higher the interference intensity of CW on the memristor. By observing the curves of the memristance with and without interference (Figure 3), it is found that the time of the memristor reaching the boundary (*t*_1_), the switching time from HRS to LRS (*t*_ON_), the switching time from LRS to HRS (*t*_OFF_), and the dynamic range of the memristor (*Ratio*) may change. In order to quantify these variations separately, four evaluation metrics corresponding to them are defined respectively, and their expressions are presented in Table 1. In the expressions, *T*_0_ denotes the period of the sinusoidal excitation voltage applied to the memristor, which is 0.01 s.

When all four evaluation metrics are equal to 0, this indicates that none of the above-mentioned four quantities of the memristor are affected after the interference. When rT_early_ > 0, it means that the memristor reaches the boundary early, and conversely, it means that the memristor delays reaching the boundary. When rT_ON_ > 0 and rT_OFF_ > 0, the memristor’s switching time is extended after the interference. |rT_OFF_ − rT_ON_| characterizes the memristor’s asymmetry. When the |rT_OFF_ − rT_ON_| equals zero, the switching time of the memristor from LRS to HRS is the same as the switching time from HRS to LRS, indicating that it is symmetrical. In this paper, we assume that the memristor maintains its original symmetry when the value is less than 1%. The larger the |rT_OFF_ − rT_ON_|, the more significant the asymmetry of the memristor. When rRatio > 0, it means that the dynamic range of the memristor increases, and the opposite means that the dynamic range of the memristor decreases.

## 3. Results and Discussion

The section below describes the optimal memristor model parameters by fitting the MMS model to experimental data of the SDC memristor, and subsequently presents and discusses simulation results of the interference effects of CWs on a memristor operating in S-Sw mode and H-Sw mode, respectively. Specifically, the influence of the operating mode of the memristor, amplitude, and frequency of CWs on the interference effects of the memristor is examined.

### 3.1. Optimal Memristor Model Parameters Based on Experimental Data

In our experiment, *V_in_* applied to the SDC memristor is a sinusoidal voltage with a peak-to-peak value of 1 V, a frequency of 5 Hz, and an offset of 0 V. To eliminate the differences in memristor responses between sinusoidal voltage cycles, we performed ten measurement cycles and captured voltage and current data for only one sinusoidal period (0.2 s) for each measurement. Based on the experimental data obtained from these ten measurement cycles, the ranges of memristor parameters (given in Table 2) were determined. After averaging these experimental data, the I-V characteristic curve of the memristor was obtained and presented as the blue line in Figure 4. Based on the experimental data, the initial memristance of the memristor M1 can be determined to be 36.45 kΩ, and the initial state variable *x*_0_ of the memristor model can be derived as 0.032 from Equation (5). After exhaustively traversing all possible parameters, the optimal parameters (listed in Table 2) were obtained when the minimum *rSSE* was 0.0101. The fitting result of the MMS memristor model with the optimal parameters to the experimental data is shown as the orange line in Figure 4.

The objective function [32] and the cost function [33] for the MMS memristor model with the optimal parameters are 9.36 × 10^−8^ and 0.01316, respectively. They are significantly smaller than the minimum values of the function reported in the literature [32] and [33], respectively. Therefore, the MMS memristor model with the optimal parameters can more accurately characterize the behavior of SDC memory compared to the VTEAM model. It can also be considered as the behavioral memristor model for the W-dopant SDC memristor and is used to examine the interference effect of CWs on it.

The memristor operates in H-Sw mode when the amplitude of the 100 Hz sinusoidal excitation voltage applied to the behavioral memristor model is ≥0.5 V and ≤1 V, and in S-Sw mode when the amplitude is ≥0.2 V and <0.5 V. The responses of the memristor in different operating modes are illustrated in Figure 5. In Section 3.2, the stimulus of the memristor operating in different switching modes without interference is the same as those depicted in Figure 5.

### 3.2. Simulation Results of Interference Effect of CWs on Memristors

In this section, the simulation results are presented separately according to the operation mode of the memristor.

#### 3.2.1. The Memristor Operating in the Hard-Switching Mode

The blue line in Figure 6 illustrates the responses of the memristor operating in H-Sw mode under 0.3 V 10 kHz CW interference. The I-V characteristics of the memristor change significantly under CW interference. The interference intensity *rSSE* of CW on the memristor is 0.74. The rRatio is 0%, which indicates that the dynamic range of the memristor is unaffected by the interference and that the memristance continues to vary between *R*_ON_ and *R*_OFF_. rT_early_, rT_ON_, and rT_OFF_ are 0%, 2.7%, 4.77%, and 7.46%, respectively, which are all greater than 0. This implies that the CW interference causes the memristor to reach its boundary earlier and results in an increase in switching time and nonlinearity. In addition, |rT_OFF_ − rT_ON_| > 1% shows an increase in the asymmetry of the memristor.

Further simulations reveal the interference effect of CWs of different frequencies and amplitudes on the memristor. The simulation results demonstrate that rRatio is constant at 0, indicating that the operating mode and dynamic range of the memristor were not affected by CWs. The simulation results of other evaluation metrics are shown in Figure 7. In general, the amplitude of CW has more significant effects on *rSSE*, rT_early_, rT_ON,_ and rT_OFF_ than the frequency of CW. According to Appendix B, we divide the frequency range of CW into low-frequency range and high-frequency range with 100 kHz as the dividing line. As is depicted in Figure 7, low-frequency (<100 kHz) CWs had more significant effects on rT_early_, rT_ON_, and rT_OFF_ than high-frequency (≥100 kHz) CWs. When Amp is fixed, frequency changes in the high-frequency range (≥100 kHz) do not affect all evaluation metrics. When Freq was fixed, *rSSE* was found to increase monotonically with increasing Amp (Figure 7a). The maximum value of *rSSE* was 2.0, obtained for a CW at a frequency of 1 kHz and an amplitude of 0.5 V.

#### 3.2.2. The Memristor Operating in the Soft-Switching Mode

In the absence of CW interference, the dynamic range of the memristor operating in the S-Sw mode does not reach the extreme value ([*R*_ON_, *R*_OFF_]), which means that the aggregation state of Ag^+^ in the conductive channel of the memristor cannot reach saturation. In contrast, under CW interference, CW superimposed on the input stimulus signal generates a stronger electric field between the memristor electrodes, which affects the accumulation and dissipation of Ag^+^ in the conductive channel, resulting in significant changes in the I-V characteristics of the memristor. Figure 8 illustrates the responses of the memristor operating in S-Sw mode when the memristor was disturbed by a 10 kHz CW with an amplitude of 0.3 V. Due to the interference of the CW, the memristor’s I-V characteristics show a greater deviation from a straight line (Figure 8a), indicating that the device’s nonlinearity was enhanced. The memristance range expanded to [*R*_ON_, *R*_OFF_] (Figure 8b), which caused the change of the operating mode of the memristor to H-Sw mode, the advancement to its boundary, and the increase of the switching time of the memristor between LRS and HRS.

Figure 9 demonstrates evaluation metrics of the interference effect of CWs on the soft-switching memristor. As shown in Figure 9a, the smaller the frequency and the higher the amplitude of CW, the higher the interference intensity of CW on the memristor. The CW at 1 kHz and 0.7 V had the highest interference intensity of 25.22 for the memristor operating in S-Sw mode. Moreover, when the Freq is fixed, *rSSE* is a monotonically increasing function of Amp. When the dynamic range of the memristor reaches its maximum and its operating mode changes to H-Sw mode (Figure 9b,c), the rRatio achieves its maximum value of 38.90%. It is apparent from Figure 9c that as the frequency decreases in the low-frequency range (<100 kHz), the amplitude range that allows the memristor operating mode to change to H-Sw mode increases, while as the frequency changes in the high-frequency range (>100 kHz), the range remains unchanged at [0.13, 0.37] V. From Figure 9d–f, we can see that when the Amp is specific, the rT_early_, rT_ON,_ and rT_OFF_ fluctuate greatly when the frequency varies in the low-frequency range (<100 kHz). In contrast, when the frequency varies in the high-frequency range (>100 kHz), the rT_early_, rT_ON,_ and rT_OFF_ are not affected by the frequency change and remain fixed.

### 3.3. Simulation Results Comparison and Discussion

In this section, the simulation results of the interference effect of CW on the memristor in Section 3.2 are compared and discussed using statistical analysis methods, and the effects of CW on the interference intensity, dynamic range, relative time variation rT, and variation of the memristor are described.

#### 3.3.1. Interference Intensity of CW on the Memristor

Figure 10 compares all interference effect evaluation metrics where the range of Amp is [0.05, 0.5] V. In Figure 10a,b the value range of each evaluation metric shows the influence of CW frequency on the evaluation metric. The larger the range, the greater the influence of CW frequency on it. The values of the red data are smaller than those of the blue data (as shown in Figure 10a,b), and the indicator ranges in the S-Sw mode are larger than those in the H-Sw mode (Figure 10c), which together indicate that the memristors operating in the H-Sw mode exhibited better anti-interference performance than those operating in the S-Sw mode. Figure 10c demonstrates the maximum rSSE values of 2.01 and 12.78 for the H-Sw and S-Sw modes, respectively, indicating that the CW can interfere with the memristor in the S-Sw mode up to 6.4 times more strongly than the memristor in the H-Sw mode. However, the memristor operating in the H-Sw mode can withstand a maximum Amp of 0.5 V (Figure 7), while the memristor operating in the S-Sw mode can withstand a maximum Amp of 0.7 V (Figure 9). Therefore, when subjected to higher amplitude CW interference, a memristor operating in H-Sw mode is more likely to burn out or undergo phase change than a memristor operating in S-Sw mode.

In addition, the interference intensity of CW on the memristor depends on the amplitude and frequency of CW. By comparing the simulation results in Figure 7 and Figure 9, it is found that the *rSSE* is larger with CW interference with high amplitude and low frequency, which indicates that the CW with high amplitude and low frequency has a more significant interference effect on the memristor. Moreover, as described in Appendix B, under a certain operating mode of the memristor and CW amplitude, the fluctuations of rT_early_, rT_ON,_ and rT_OFF_ become larger as the CW frequency decreases in the low-frequency range (<100 kHz). In contrast, when the frequency is varied in the high-frequency range (>100 kHz), the above three evaluation indicators remain almost constant. This suggests that when the CW amplitude is certain, the interference effect on the memristor is more significant for CWs with lower frequencies and almost uniform for CWs with high frequency. According to the fingerprint characteristic of the memristor [38], the uniformity is due to the fact that the hysteresis line of the memristor shrinks to a single-valued function when the memristor is driven by the high-frequency CW alone. Therefore, the intensity of the interference effect of high-frequency (>100 kHz) CW on the memristor is mainly related to the amplitude of CW.

#### 3.3.2. Interference Effects of CW on the Dynamic Range of Memristors

Since rRatio is constant at 0 in the hard mode and rRatio has a higher value in the soft mode (as shown in Figure 10c), CW interference does not change the dynamic range of the memristor operating in H-Sw mode, but significantly increases the dynamic range of the memristor operating in soft-switching mode. This is because the aggregation state of Ag^+^ in the conductive channel of the memristor working in hard switching mode could reach saturation; that is, its dynamic range had reached a limit ([*R*_ON_, *R*_OFF_]), while the memristor working in soft switching mode could not. Therefore, the additional superimposed CW of appropriate magnitude will continue to drive Ag^+^ aggregation in the conductive channel until the aggregation state is saturated, resulting in a smaller r memristance of the LRS of the memristor and an increase in the dynamic range of the memristor. Moreover, the increased dynamic range causes the I-V characteristics of the memristor to deviate more from the straight line, which in turn leads to an increase in the nonlinearity of the memristor. Thus, the memristance of the LRS will be smaller, and the dynamic range of the memristor will be increased. Moreover, the increased dynamic range will cause the I-V characteristics of the memristor to deviate more from the straight line, which in turn leads to an increase in the nonlinearity of the memristor.

#### 3.3.3. Interference Effect of CW on Relative Time Variation rT of Memristors

The interference effect of CW on the four relative time variations of rT_early_, rT_OFF_, rT_ON,_ and|rT_OFF_ − rT_ON_| determines the interference effects of CW on the arrival time at the boundary, switching time, and symmetry of the memristor. Comparing the percentages of various cases in hard and soft switching modes (given in Table 3), CW interference generally prolonged the switching time of the memristor operating in H-Sw mode since the percentages of both rT_ON_ > 0 and rT_OFF_ > 0 in H-Sw mode exceeded 98%. Another interesting finding is that the percentage of rT_OFF_ > 0 exceeded 99% for both switching modes and was greater than the percentage of rT_ON_ > 0. The reason for this finding could be closely related to the fact that the switching threshold voltage of the OFF state of the memristor is smaller than that of the ON state. In addition, since the dynamic range of the memristor operating in S-Sw mode increased after CW interference, the percentages of rT_early_ > 0 and |rT_OFF_ − rT_ON_| > 1% were both higher in the S-Sw mode than those in the H-Sw mode. In other words, under CW interference, the memristor operating in the S-Sw mode is more likely to reach the boundary in advance and has worse symmetry than the memristor operating in the H-Sw mode.

The distribution of the frequency and amplitude of CW for rT_early_ < 0 and |rT_OFF_ − rT_ON_| ≤ 1% is shown in Figure 11. As shown by the purple symbols in Figure 11a, when the memristor is operating in hard-switching mode, CW with an amplitude higher than 0.45 V or lower than 0.25 V may delay the memristor from reaching the boundary, while for the memristor operating in soft-switching mode, the delay of the memristor to reach the boundary generally occurs for CW disturbances with amplitude less than 0.1 V and frequency less than 6.1 kHz (as shown by the orange symbols in Figure 11a). Figure 11b illustrates that low-amplitude CW interference generally does not cause a change in the symmetry of the memristor. For the H-Sw mode, the CW amplitude that did not cause a change in the symmetry of the memristor was generally less than 0.18 V, while for the S-Sw mode, the value was less than 0.06 V.

#### 3.3.4. Interference Effect of CW on the Variation of Memristance

Under CW interference, the variation of memristance does not depend on the operating mode of the memristor. However, it is proportional to the amplitude of CW and inversely proportional to the frequency of CW. Since the studied memristor is voltage-controlled, it is easy to understand that the variation in memristance is proportional to the amplitude of CW. As shown in Figure 12, this section illustrates the interference effect of CW frequency on the variation of memristance, using a memristor operating in H-Sw mode as an example. It can be clearly seen from Figure 12 that a 5 kHz CW caused more variation than a 10 kHz CW when the memristor was disturbed by a 0.3 V CW. In addition, at a certain CW amplitude, the variation of the memristance during the memristor switch from LRS to HRS is greater than during the memristor switch from HRS to LRS.

In summary, the memristors operating in H-Sw mode have better immunity to CW interference than those operating in S-Sw mode. The CW may affect the switching time, nonlinearity, symmetry, time to the boundary, and variation of the memristance, regardless of the operating mode of the memristor, and the CW can even have a considerable impact on the dynamic range of the memristor operating in S-Sw mode. It is well known that bipolar memristors are highly appropriate basic devices for implementing synaptic functions and have been widely used in neuromorphic computing systems [39]. In general, to reduce the effects of CW interference on the reliability of neuromorphic computing systems, memristors operating in H-Sw mode should be preferred.

In addition, we can further speculate on some possible effects of the aforementioned CW interference effects on memristors on the reliability of memristive neuromorphic computing systems. If the CW interference occurs only in the weight processing phase, early arrival at the boundary or an increased variation may cause deviations in the read memristance, resulting in incorrect weights. According to the literature [7], CW interference causes an increase in the asymmetry of the memristors, which may lead to a significant decrease in the classification accuracy of the neuromorphic system. It has been demonstrated that a large dynamic range can bring high precision and weight mapping ability to neuromorphic systems, and inconsistencies in dynamic range, nonlinearity, and symmetry among devices can lead to poor convergence rates during the training process of neuromorphic systems [24]. According to the above interference effects, the large dynamic range and the nonlinearity and symmetry inconsistency between devices may coexist for memristors operating in S-Sw mode subjected to the CW interference. Hence, their final impact on the accuracy of neuromorphic systems is still difficult to identify. Finally, we suggest that the effect of CW interference on memristors in memristor-based neuromorphic systems should be taken into account, and it is essential to investigate the appropriate protective solutions.

We must accept that the differences in materials, device architectures, and physical methods utilized to develop and fabricate the memristors result in highly diverse characteristics. However, we hypothesize that their EMI effects may be similar. This research demonstrates an exemplary approach to investigating the impact of other EMI signals on various types of memristors. Based on this method, we are currently conducting some simulation and experimental studies on the interference of electromagnetic noise and electromagnetic pulse on the Knowm memristors. Using Gaussian white noise to simulate electromagnetic noise, Gaussian white noise is superimposed on the input of the memristor. It is found that the interference effect of Gaussian white noise on the I-V characteristics of the memristor is closely related to the intensity of Gaussian white noise, and this experimental result is consistent with the simulation results. In addition, we also simulated the interference effect of electromagnetic pulse on the memristor, and the simulation results show that the interference effect of the electromagnetic pulse depends on the amplitude, pulse width, and pulse interval of the electromagnetic pulse. More specific impact regularities will require extensive traversal of the parameters of the EMI signal to be concluded. The results will be reported in an upcoming article.

## 4. Conclusions

This paper establishes an optimal memristor model with the Knowm SDC memristor as an example. Based on this model, we have provided a deeper insight into the interference effect of CWs on memristors driven sinusoidally, as well as a method for investigating the electromagnetic interference effects of memristors. Simulation results indicate that CW may affect the switching time, dynamic range, nonlinearity, symmetry, time to the boundary, and the variation of the memristance of the memristor. Moreover, the specific interference effect depends on the operating mode of the memristor, as well as the amplitude and frequency of the CW.

The comparative investigation has demonstrated the specific interference effects of CW on the interference intensity, dynamic range, relative time variations rT, and the variation of memristance. First, memristors operating in H-Sw mode have stronger CW immunity than those operating in S-Sw mode but are at greater risk of burnout and phase change. In the same operating mode, changing the amplitude of the CW has a higher impact on the interference effect of the memristor than changing its frequency, and at a specified frequency of CW, the interference intensity of CW on the memristor is proportional to its amplitude. Second, the dynamic range of a memristor operating in H-Sw mode is unaffected, whereas the dynamic range of a memristor working in S-Sw mode is significantly increased, resulting in a significant increase in nonlinearity and asymmetry. Third, under CW interference, the memristor operating in the S-Sw mode is more likely to reach the boundary in advance and has worse symmetry than the memristor operating in the H-Sw mode. The disturbing of the symmetry of the memristor typically occurs when the CW amplitude is relatively large. Four, the variation of memristance does not depend on the operating mode of the memristor but is proportional to the amplitude of CW and inversely proportional to the frequency of CW.

Finally, this paper points out that the memristor interference effect arising from the CW interference will have a significant impact on the reliability of memristor-based neuromorphic circuits and should be taken seriously. The present study lays the groundwork for the EMI effect assessment and electromagnetic protection design of memristive neuromorphic systems.

## Figures and Tables

**Figure 1 sensors-22-05785-f001:**
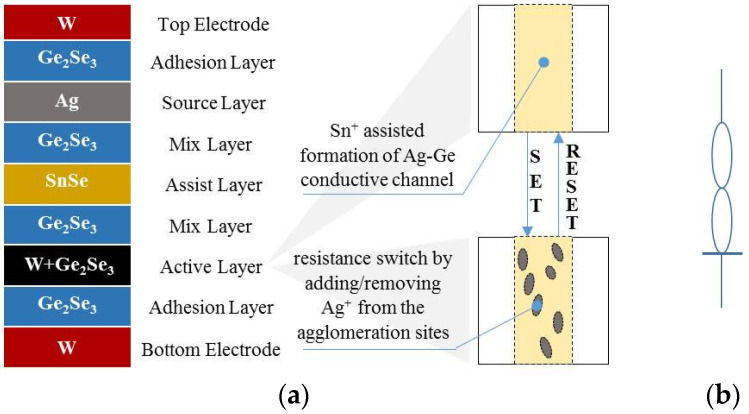
The W-dopant SDC memristor [31]. (**a**) Materials (**left**) and a graphical representation of the switching mechanism (**right**); (**b**) symbol of the SDC memristor.

**Figure 2 sensors-22-05785-f002:**
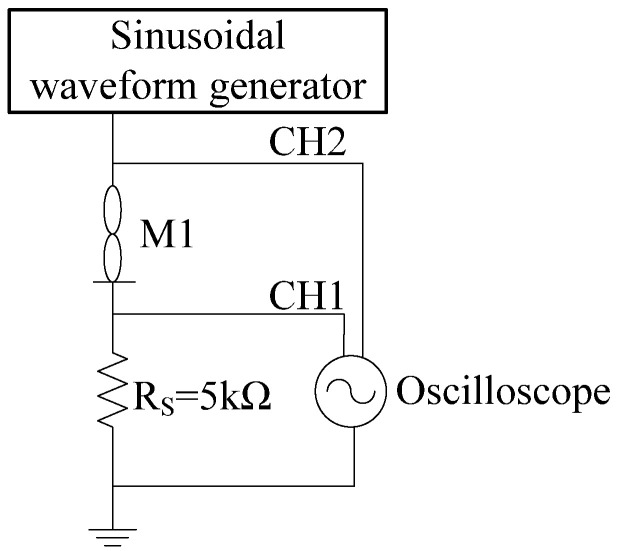
The experimental connection setup.

**Figure 3 sensors-22-05785-f003:**
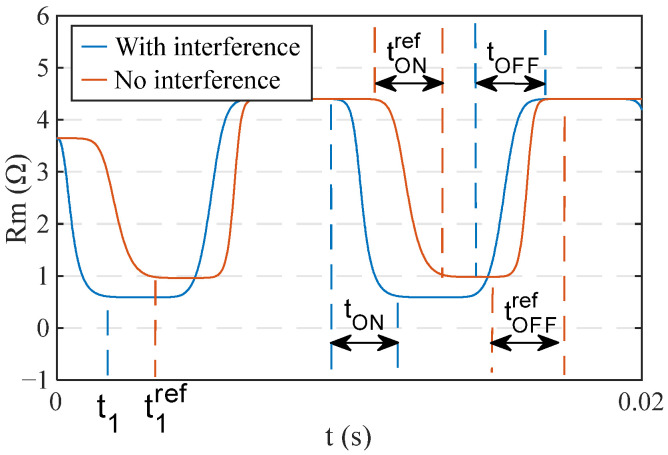
Diagram of the memristance of the memristor with and without interference.

**Figure 4 sensors-22-05785-f004:**
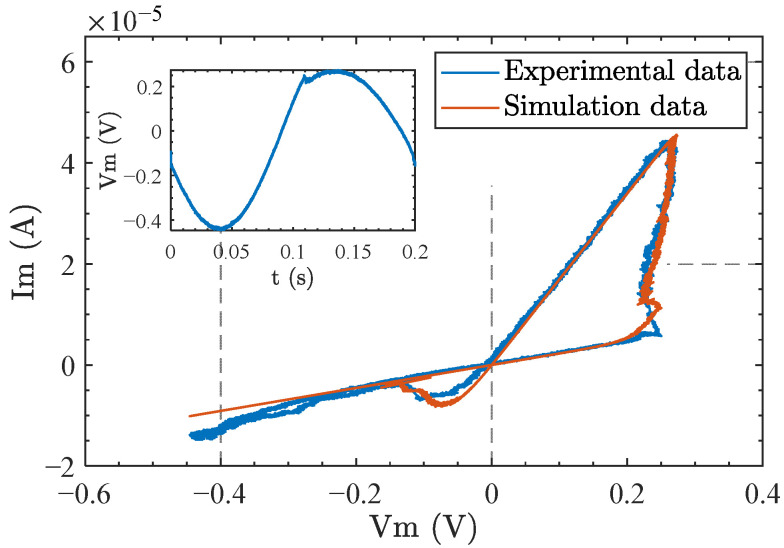
The MMS memristor model with the optimal parameters fits the experimental data of the SDC memristor. The applied voltage across the memristor is shown in the subwindow.

**Figure 5 sensors-22-05785-f005:**
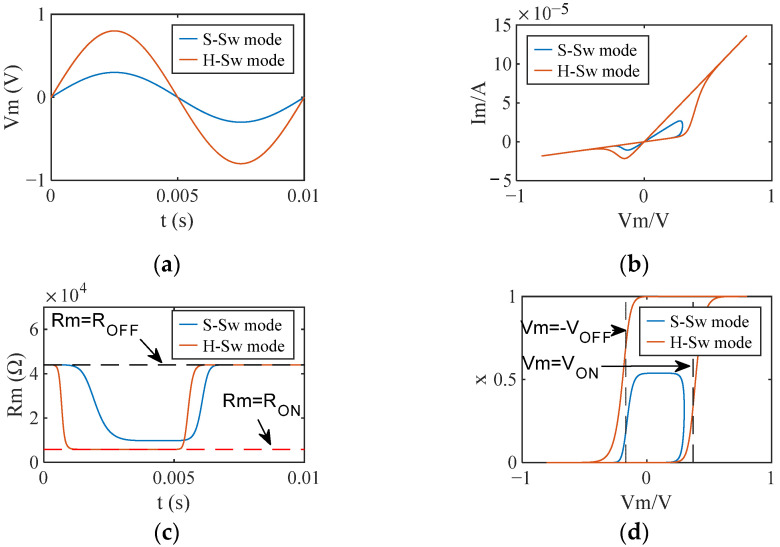
Responses of the memristor in different operating modes. (**a**) The sinusoidal excitation voltage with a frequency of 100 Hz. Sinusoidal excitation voltage with an amplitude of 0.5 V enables the memristor to operate in H-Sw mode; sinusoidal excitation voltage with an amplitude of 0.3 V lets the memristor operate in S-Sw mode; (**b**) I-V characteristics; (**c**) memristance; (**d**) state variable.

**Figure 6 sensors-22-05785-f006:**
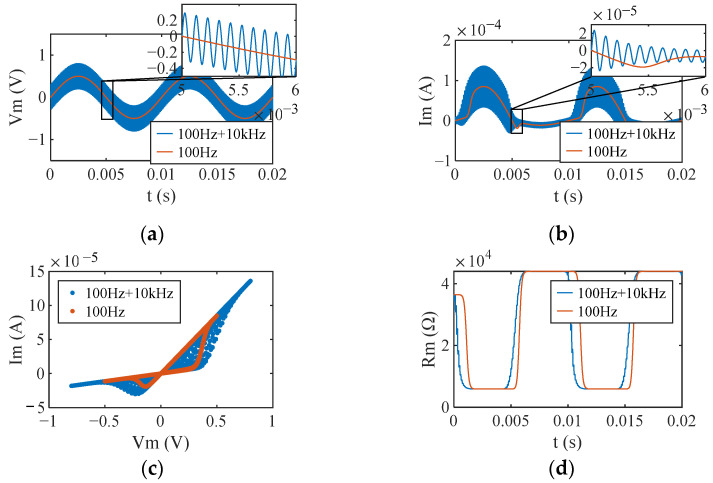
Responses of a memristor operating in hard-switch mode with (blue line) and without interference (orange line) from the CW at 0.3 V and 10 kHz. (**a**) Applied voltages; (**b**) currents at the memristor; (**c**) I-V characteristics; (**d**) memristance.

**Figure 7 sensors-22-05785-f007:**
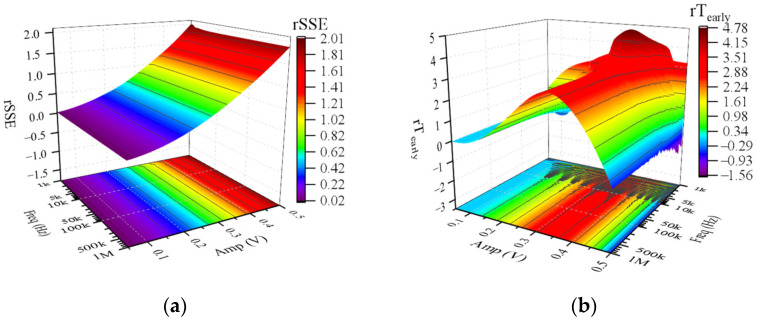
Evaluation metrics of the interference effect of CWs on the hard-switching memristor. (**a**) *rSSE*; (**b**) rT_early_; (**c**) rT_ON_; (**d**) rT_OFF_.

**Figure 8 sensors-22-05785-f008:**
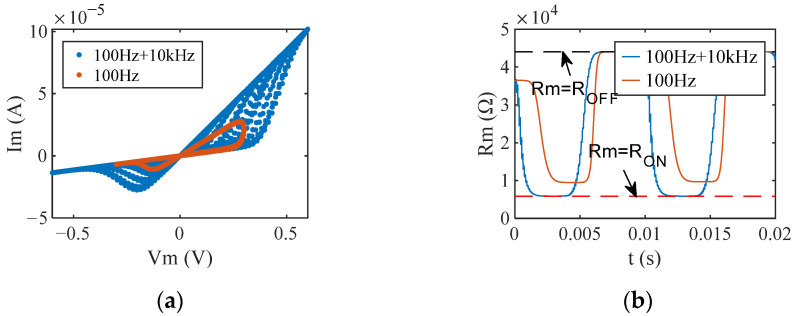
The responses of a memristor operating in soft-switching mode with (blue line) and without interference (orange line) from the CW at 0.3 V and 10 kHz. (**a**) I-V characteristics; (**b**) memristance.

**Figure 9 sensors-22-05785-f009:**
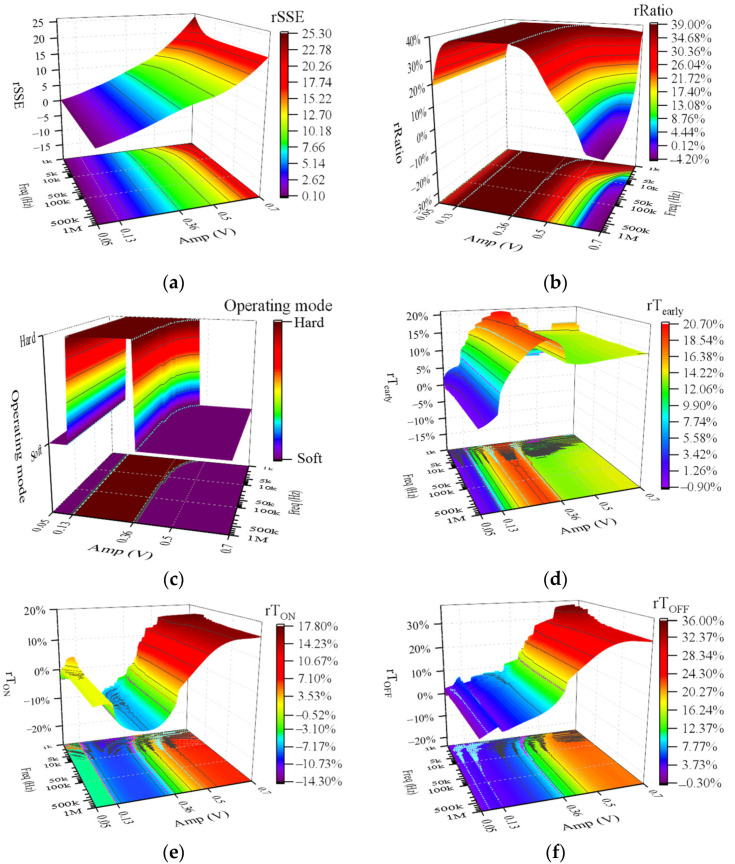
Evaluation metrics of the interference effect of CWs on the soft-switching memristor. (**a**) *rSSE*; (**b**) rRatio; (**c**) operating mode; (**d**) rT_early_; (**e**) rT_ON_; (**f**) rT_OFF_.

**Figure 10 sensors-22-05785-f010:**
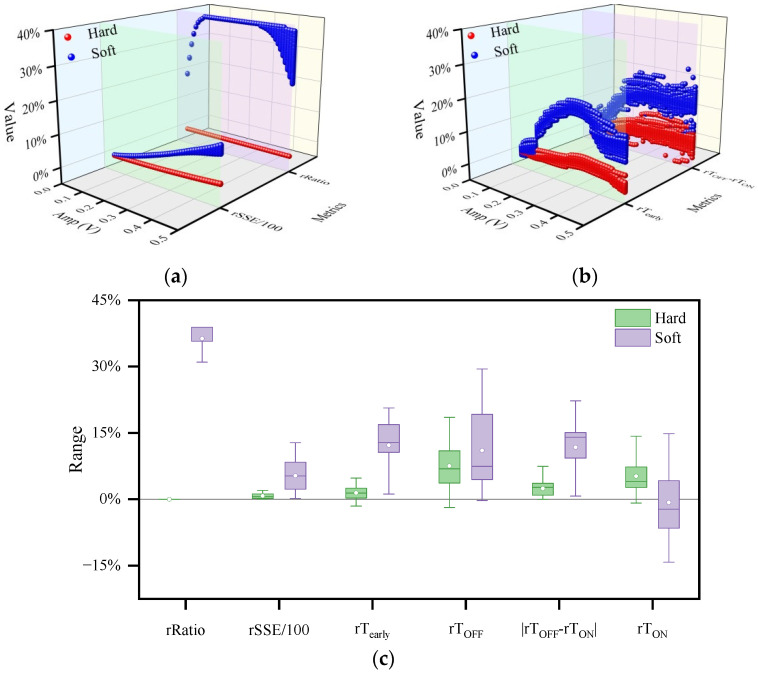
Comparison of evaluation metrics of the memristor under CW interference. (**a**) *rSSE*/100 and rRatio with Amp; (**b**) rTearly and rT_OFF_ − rT_OFF_ with Amp; (**c**) ranges of rRatio, *rSSE*/100, rT_early_, rT_OFF_, |rT_OFF_ − rT_ON_|, and rT_ON_. To enable *rSSE* and rRatio to share the vertical percentage coordinate, all *rSSE* values are divided by 100; **Soft** indicates that the memristor operates in S-Sw mode, whereas **Hard** indicates that the memristor operates in H-Sw mode.

**Figure 11 sensors-22-05785-f011:**
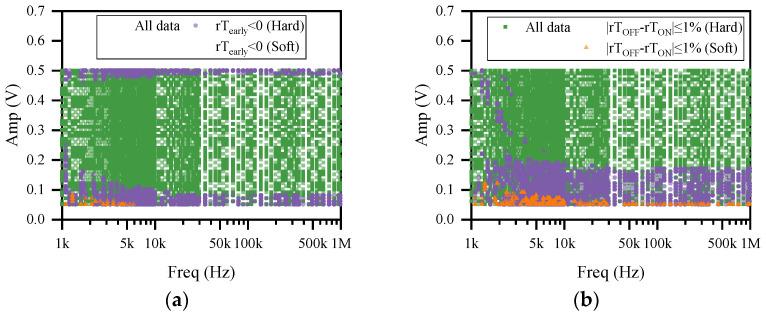
The distribution of CW frequencies and amplitudes for the following cases: (**a**) rT_early_ < 0 (**b**) |rT_OFF_ − rT_ON_| ≤ 1%.

**Figure 12 sensors-22-05785-f012:**
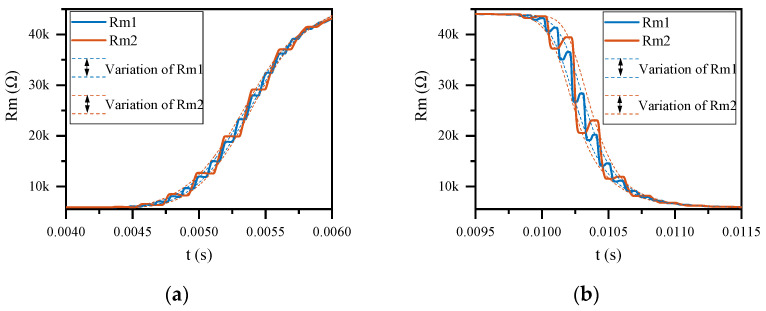
The variation of memristance when the memristor operating in H-Sw mode is disturbed by 0.3 V CW. Rm1 and Rm2 are the memristances at 10 kHz and 5 kHz CW interference, respectively. (**a**) The variation of the memristance during the memristor switch from LRS to HRS; (**b**) the variation of the memristance during the memristor switch from HRS to LRS.

**Table 1 sensors-22-05785-t001:** Expressions and descriptions of the metrics used to quantify the EMI effects.

Metrics	Expressions	Description
*rSSE*	$\frac{\sum_{i=1}^{n}{\left(V_{with\;intference,i}-V_{ref,i}\right)}^{2}}{\sum_{i=1}^{n}V_{ref,i}^{2}}+\frac{\sum_{i=1}^{n}{\left(I_{with\;intference,i}-I_{ref,i}\right)}^{2}}{\sum_{i=1}^{n}I_{ref,i}^{2}}$	Interference intensity of CW on memristors
rT_early_	$[(t_{1}^{ref}-t_{1})/T_{0}]\times 100\%$	The relative variation of the time of the memristor reaching the boundary
rT_ON_	$[(t_{\mathrm{ON}}-t_{\mathrm{ON}}^{ref})/T_{0}]\times 100\%$	The relative variation of the switching time from HRS to LRS
rT_OFF_	$[(t_{\mathrm{OFF}}-t_{\mathrm{OFF}}^{ref})/T_{0}]\times 100\%$	The relative variation of the switching time from LRS to HRS
rRatio	$[(\frac{R_{\max}}{R_{\min}}-\frac{R_{\max}^{ref}}{R_{\min}^{ref}})/\frac{R_{\mathrm{OFF}}}{R_{\mathrm{ON}}}]\times 100\%$ ^1^	The relative variation of dynamic range

^1^ *R_max_* and *R_min_* represent the maximum and minimum memristance of the memristor with interference, respectively. *Ratio* = *R*_max_/*R*_min_ is the dynamic range of the memristor. $R_{\max}^{ref}$ and $R_{\min}^{ref}$ represent the maximum and minimum memristance of the memristor without interference, respectively.

**Table 2 sensors-22-05785-t002:** Parameter ranges and step sizes for traversal and the optimal parameters of the memristor model.

	*R*_ON_/kΩ	*R*_OFF_/kΩ	*V*_ON_/V	*V*_OFF_/V
Parameter ranges	[5.8, 5.9]	[44, 44.1]	[0.3, 0.4]	[0.1, 0.2]
Step sizes	0.01	0.01	0.01	0.01
Optimal parameters	5.88	44.02	0.37	0.17

**Table 3 sensors-22-05785-t003:** According to the simulation data in Section 3.2, the percentages of various cases in the two switching modes.

Cases	H-Sw Mode	S-Sw Mode
rT_ON_ > 0	99.22%	45.01%
rT_OFF_ > 0	99.42%	99.41%
rT_ON_ > 0 and rT_OFF_ > 0	98.8%	44.98%
rT_early_ > 0	87.23%	99.55%
|rT_OFF_ − rT_ON_| > 1%	73.85%	97.39%

## Data Availability

Not applicable.

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
