# Peer review of "Electromagnetic Interference Effects of Continuous Waves on Memristors: A Simulation Study"

_sensors, 2022, doi:10.3390/s22155785_

Round 1
Reviewer 1 Report
The manuscript explains the effect of continuous waves on the memristor. As memristor is considered an emerging candidate for next-generation memory devices, this study is inevitable for the practical application of memristor in electronic devices. Although the manuscript is well written with a unique idea, I have only a minor comment:
How will b EMI affect irregular and discontinuous was on the memristors?
Reviewer 2 Report
I have read manuscript entitled “Electromagnetic interference effects of continuous waves on memristors: A simulation study” in a fairly detailed fashion. This paper deals with the optimal memristor model for a commercially available memristor, which provides a deeper insight into the interference effect of continuous waves on memristors driven sinusoidally, as well as a method for investigating the electromagnetic interference effects on memristors. In general, the topic of this manuscript is academically and technologically relevant. However, there exist some points that need to be clarified.
In which software was the MMS memristor model implemented? The authors should make the program code available either in a paper appendix or supplementary material!
At what frequency of the applied voltage was the current-voltage characteristic of memristor (given in Fig. 4) measured? Based on how many measurement cycles were the ranges of memristor parameters (given in Table 2) determined?
Why was it chosen that the CW frequency of 100 kHz is critical for the division into low-frequency and high-frequency ranges?
The results shown in Figure 8 are not explained from the point of view of the memristor. Why does the current-voltage characteristic change?
A careful revision of English is also recommended.
Generally, the paper is interesting, but also a little confusingly written, especially in the Section 3.3. Based on the above, I think that manuscript in this version cannot be accepted for publication in Sensors since it needs a major revision.
Reviewer 3 Report
This paper presents simulation work about electromagnetic interference (EMI) effects of continuous waves on memristor devices. I recommend publication because this paper is well written. I cannot find critical technical issues and error. Below comments are optional (not mendatory)
1) Methods about simulation can be more detailed
2) Memristor’s history can be added more as well as Chua’s one. For example HP
3) Applications can be added for memristor more such as neuromorphic.
4) Comparison is need using other types devices for EMI. Need to specify the spec. Memristor can has several type. PRAM. MRAM. And RRAM has many types interface and CBRAM, an so on…They has all different characteristics such as switching curve and speed. Need to justify why you use memristor.
Round 2
Reviewer 2 Report
I suggest accepting the manuscript in present form for publication in Sensors.